# A transthalamic pathway crucial for perception

Christina Mo [1,2] ✉, Claire McKinnon[2] & S. Murray Sherman [2] ✉

Perception is largely supported by cortical processing that involves communication among multiple areas, typically starting with primary sensory cortex and then involving higher order cortices. This communication is served in part by transthalamic (cortico-thalamo-cortical) pathways, which ubiquitously parallel direct corticocortical pathways, but their role in sensory processing has largely remained unexplored. Here, we suggest that transthalamic processing propagates task-relevant information required for correct sensory decisions. Using optogenetics, we specifically inhibited the pathway at its synapse in higher order somatosensory thalamus of mice performing a texture-based discrimination task. We concurrently monitored the cellular effects of inhibition in primary or secondary cortex using two-photon calcium imaging. Inhibition severely impaired performance despite intact direct corticocortical projections, thus challenging the purely corticocentric map of perception. Interestingly, the inhibition did not reduce overall cell responsiveness to texture stimulation in somatosensory cortex, but rather disrupted the texture selectivity of cells, a discriminability that develops over task learning. This discriminability was more disrupted in the secondary than primary somatosensory cortex, emphasizing the feedforward influence of the transthalamic route. Transthalamic pathways may therefore act to deliver performance-relevant information to higher order cortex and are underappreciated hierarchical pathways in perceptual decision-making.

The conventional view of sensory processing in the cortex, as defined by textbook accounts, is that information is processed in a hierarchical fashion starting with primary sensory areas to secondary areas, etc. up the ladder[1–4]. In all of these accounts, the processing of information between cortical areas involves only direct connections. However, increasing evidence indicates that information from the primary to secondary cortex can arrive either directly or indirectly via higher-order thalamic nuclei (Fig. 1A). These feedforward cortico-thalamo-cortical, or transthalamic, pathways often if not always are present in parallel to direct pathways and use 'driver' type synapses that support the fast, robust propagation of stimulus information[5–9]. Feedforward transthalamic pathways are thus well positioned to influence higher-order processing.

There is some evidence to suggest that these indirect pathways are surprisingly powerful inputs to the higher-order cortex[10] and carry distinct, task-relevant information, compared to the direct cortical projections[11–14]. For example, in mice moving through a visual environment, it is the thalamocortical input rather than the corticocortical input that has a stronger influence on the response patterns of the higher-order cortex[12]. During a whisker-based perceptual task, activating the transthalamic-projecting cells in the primary somatosensory cortex enhanced whisker-based detection, but activating the corticocortical-projecting cells had no behavioral impact[14]. Transthalamic pathways appear to be key contributors to cortical processing, but it is not clear what information propagates through this pathway to the higher-order cortex. Here we show that the projection from S1

[1]The Florey Institute of Neuroscience and Mental Health, University of Melbourne, Melbourne, Victoria, Australia. [2]Department of Neurobiology, University of Chicago, Chicago, Illinois, USA. ✉e-mail: christina.mo@florey.edu.au; msherman@bsd.uchicago.edu

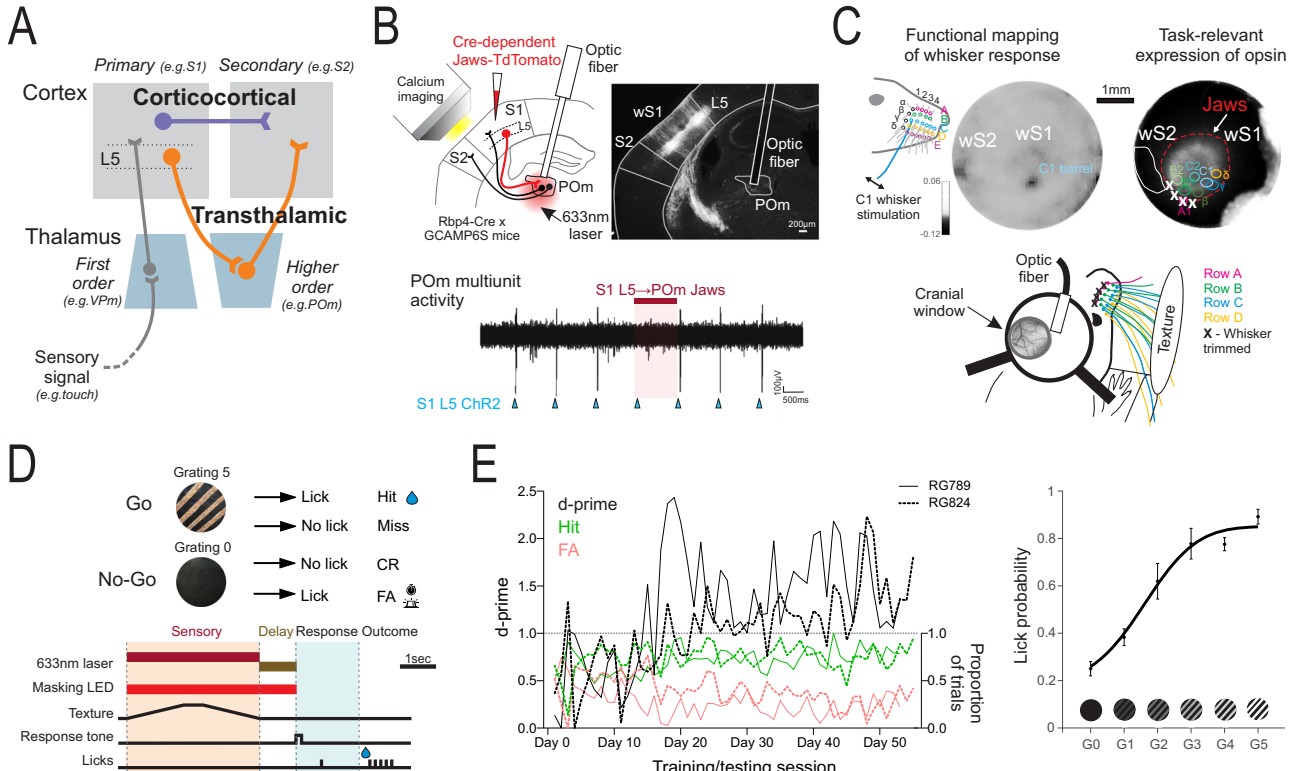

**Fig. 1 | Targeting the somatosensory transthalamic pathway in a discrimination task. A** Feedforward transthalamic pathways from layer 5 of the primary cortex parallel corticocortical projections in sensory processing. **B** (Top left) Expression of the Jaws inhibitory opsin in S1 layer 5 cells and (top right) targeted inhibition via optic fiber implant in anterior POm (see Supplementary Fig. 1). The cellular impact was monitored in S1 or S2 by calcium imaging. (Bottom) Example trial of electrophysiological recording in POm showing activation of S1 L5 suppressed by Jaws in POm terminals (see Supplementary Fig. 2). **C** (Top left) Functional intrinsic signal optical imaging of whisker-stimulated responses in S1 and S2, viewed through a cranial window. (Top right) For each mouse, areas of S1 that represent individual whiskers ('barrels') were mapped and overlaid with the Jaws-TdTomato expression. (Bottom) Any barrels that did not localize with Jaws expression had their corresponding whiskers trimmed (Row A in the example

shown; Row E is not shown for clarity). wS1: whisker S1, wS2: whisker S2. **D** (Top) The texture discrimination task followed a go/no-go design where licking to a texture with gratings (G5) was rewarded and licking to a texture of smooth foil (G0) was punished. (Bottom) The task had sensory stimulus, delay, and response epochs. For laser inhibition trials, a 633 nm laser was activated during either the sensory period or the delay period of the task. **E** (Left) Performance of two example mice during G5 vs G0 discrimination. (Right) Psychometric performance to various textures during no-laser trials from an example mouse (n = 7 sessions). The G5 texture had gratings made of P20 grit, G4 texture made of P150, G3 of P220, G2 of P1500, G1 of foil strips, and G0 is foil only. Error bars show the standard error of the mean. Source data are provided as a Source Data file. Brain sections traced from the Allen Mouse Brain Atlas, https://atlas.brain-map.org/[58].

layer 5 to the posterior medial nucleus of the thalamus (L5 to POm) is essential for discrimination and correct representation of stimulus features in S2.

We tested the function of the transthalamic pathway in the whisker system of the mouse by using optogenetic inhibition of layer 5 (L5) terminals in the higher-order thalamus from the primary somatosensory cortex (S1). We observed the effects of this inhibition on the animals' ability to perform a whisker-based discrimination task that relies on cortical communication[15,16]. At the same time, neuronal responses in S1 and secondary somatosensory cortex (S2) were monitored using 2-photon calcium imaging. Importantly, instead of suppressing the pathway during the entire trial[14,15,17], which limits interpretation regarding perceptual processing, we separately inhibited during the period of sensory whisking and a delay epoch separating sensation and motor response. We found that texture discrimination performance was impaired by inhibition during both the sensory and delay periods, and inhibition during the sensory epoch also disrupted cell selectivity for the rewarded texture in S2, with smaller effects on cells in S1. Transthalamic pathways thus appear critical in delivering performance-relevant information to the higher-order cortex and are powerful but underappreciated hierarchical pathways in perception.

## Results

### Targeted inhibition of the first leg of the transthalamic pathway

We selectively inhibited the first leg of the transthalamic pathway from S1 L5 by injecting a Cre-dependent adeno-associated virus (AAV) carrying the inhibitory opsin, Jaws, into S1 of transgenic mice expressing Cre in layer 5 cells (Rbp4-Cre). An optic fiber was implanted in the higher-order thalamic nucleus, POm, to suppress terminals from S1 L5 (Fig. 1B). The implant was positioned in the anterior-dorsal part of POm, which receives terminals from S1 L5 and contains cells that project to S2 (Supplementary Fig. 1), and also cells that project to S1[5]. We verified Jaws inhibition at these terminals in separate electrophysiology experiments (Fig. 1B, Supplementary Fig. 2). The impact of inhibition on cortical cells was monitored, thus allowing functional assessment of inhibiting the transthalamic pathway from S1 L5 to POm to S2 or to S1 (Fig. 1B).

To locate S1 and S2, we mapped hemodynamic changes to stimulation of the whiskers using intrinsic signal optical imaging of the cortex through a cranial glass window (Fig. 1C). By overlaying the images of whisker response maps and expression of Jaws-TdTomato in S1 L5, we could exclude any mice with expression that had spread to S2. The overlay also allowed the identification of any S1 barrels that did not localize with the expression of Jaws, the corresponding whiskers of which were trimmed. Whiskers that did not contact the texture were

left intact. Thus, this trimming reduced the sensory input from whiskers that could not be controlled by Jaws.

### Inhibition of S1 layer 5 to POm terminals impairs discrimination performance

To test the behavioral effect of inhibiting the S1 L5 to POm projection, we used a texture discrimination task that requires S1 to S2 corticocortical communication[15,18]. Textures were presented to the whisker fields of water-restricted, head-fixed mice that indicated their discrimination choice by a lick or no-lick response in a go/no-go task design (Fig. 1D). A lick response to a texture with a coarse grating (Grating 5, G5) was rewarded with a drop of water (hit). A lick response to a smooth texture made of foil (Grating 0, G0) was punished with an alarm and 12 s timeout (False alarm). No-lick responses to G5 were regarded as a miss, whilst no-lick to G0 was a Correct Rejection (CR). A delay of 1 s separated the texture presentation 'sensory period' of the task from the 'response period' and subsequent 633 nm laser to activate Jaws inhibition could be applied during the sensory or delay epochs. Mice generally required 3–4 weeks to successfully discriminate G5 from G0, defined as a d-prime performance above 1 for 2 consecutive days (Fig. 1E). Psychometric performance was then probed by presenting various gratings of lower coarseness (G4–G1).

S1 L5 to POm terminals were inhibited with a 633 nm laser measured $4 \pm 0.2$ mW at the tip of the implant in POm (-127 mW/mm$^2$) (Fig. 2A). Inhibition during the texture presentation period of the task ("sensory laser") caused higher error rates compared to trials without laser activation ("no-laser") (Fig. 2B) Psychometric testing on a range of textures revealed that mice were more likely to lick to the coarser gratings (G3–G5) and less likely to lick to the non-rewarded smooth texture (G0) (Fig. 2C, top, black curve). Sensory laser application resulted in a compressed psychometric function (Fig. 2C, red curve). Quantification of psychometric curves also reflected an increase in guess rates (false alarm) and lapse rates (miss) ($p = 0.0078$ and $p = 0.0039$, respectively, see Supplementary Table 1 for additional statistics). The steepness of the psychometric curve (sensitivity) was reduced ($p = 0.0039$) but the discrimination threshold (bias) was not shifted ($p = 0.92$, Supplementary Table 1). As another index of performance, we calculated the average d-prime for each grating texture compared to the G0 texture. During sensory laser trials, d-prime was reduced for all textures compared to no-laser trials (Fig. 2C, bottom). The large effect size of the deficit during sensory laser application was clear when compared to trials performed in the absence of whiskers (Fig. 2D).

On separate days, the same mice were also tested on the effects of laser application during the delay period of the task (Fig. 2E). This caused a trend to increase error rate (Fig. 2E, bottom) and significantly shifted the psychometric curve to the right (Fig. 2F, top) with an expected increase in bias (threshold), but no effect on the slope of the curve ($p = 0.039$ and $p = 0.91$ respectively, Supplementary Table 1). D-prime performance was reduced during delay laser trials for the textures difficult to discriminate against G0 (G1, G2) but not for those more easily discriminated (G3, G5) (Fig. 2F, bottom). Thus, the effect of the laser on performance was significant when applied during the delay epoch but more subtle compared to the laser during the sensory period (Fig. 2D).

Red light activation in the brain by itself can potentially have effects on mouse behavior through activation of retinal circuits[19,20]. Thus, in addition to a masking LED (Fig. 1D), we performed two controls: One control was to inject mice with AAV carrying TdTomato without Jaws (n = 4). These control mice did not show differences between no-laser and sensory laser trials for d-prime, error rate, or psychometric curve parameters (Supplementary Fig. 3A, B). The same TdTomato-expressing mice were also unaffected by laser application during the delay epoch of the task (Supplementary Fig. 3C, D). The

TdTomato control group showed that red laser delivery alone did not contribute to the behavioral effects seen in Jaws-expressing mice.

As a second, within-mouse control, 3 of the 9 Jaws-expressing mice expert in the discrimination task were implanted with a second optic fiber, anterior to POm, well away from Jaws-expressing terminals (Fig. 2G). The mice were then tested under both sensory and delay laser conditions within the same session using a subset of textures (G0, G1, G2, G5). When the laser was delivered through the POm implant, the effects of sensory and delay laser were replicated (Fig. 2G, top right). When the laser was attached to the anterior control implant, there was no effect of laser application on performance (Fig. 2G, bottom right). In further support that our laser effects were Jaws-mediated, there was a positive correlation between the errors made during sensory laser (guess and lapse rates) and the estimated level of Jaws activation in POm (Fig. 2H).

A potential explanation for the impaired ability to discriminate textures during sensory laser could be reduced sensory sampling, that is, reduced whisking of the textures. However, whisker analysis during the sensory period did not show differences in the whisking rate or amplitude between no-laser and sensory laser trials (Fig. 2I, Supplementary Fig. 4). In addition, the texture is moved within 2 cm of the whisker pad, allowing for 'passive' texture sampling. Reduced ability to whisk is thus an unlikely explanation of impaired behavioral performance.

### Pathway inhibition reduces performance on a detection task

In contrast to discrimination, the ability to detect a whisker-based stimulus has been shown to be independent of S1[15,21] and, as an extension, independent of the S1 L5 transthalamic projection. We tested this hypothesis with the same Jaws-mediated inhibition on a detection task analogous in design to our discrimination task (Fig. 3A). Instead of a whisker deflection task with overlapping stimulus and response windows[14], we trained mice to detect the deflection of a panel sitting in their whisker field, which was then followed by a 1 s delay before a response cue (Fig. 3B). Catch trials of no-movement were presented to test for conditioned licking to the response cue. We trained 4 Jaws-injected mice in the detection task, 3 of which also learned the discrimination task. Inhibition of the S1 L5 to POm projection shifted the psychometric curve to the right (Fig. 3C), increasing the threshold of detection but not affecting sensitivity, lapse and guess rates (Fig. 3D). Controls consisted of TdTomato-injected mice (n = 3, 1 from discrimination task) and Jaws-injected mice with the laser attached to a second optic fiber implant, as described in the discrimination task (n = 2) (Fig. 2H). Detection performance in these control mice was no different in laser trials (Fig. 3E, F). The impaired detection performance corroborates a study that used inhibitory DREADDs to suppress the projection[14]. However, in contrast to Takahashi et al., we did not find a reduction in response to no-movement catch trials (Fig. 3C), which could indicate inhibited response from the use of DREADDs in general. This is perhaps because optogenetics can be selectively applied to the sensory and delay epochs, avoiding inhibition during the entire trial, including the response period. The transthalamic pathway thus acts to reduce the performance threshold, rather than be essential for somatosensory detection.

### Suppressing the pathway disrupts texture selectivity in cortical cells

We focused our cortical cell analyses on the discrimination task and the sensory period, the behavioral epoch during which we saw the largest effect of transthalamic inhibition (Fig. 2B, C). The behavioral impairment resulting from inhibition of S1 L5 to POm terminals could affect encoding in any areas targeted by POm involved in texture discrimination, notably S1 and S2[5,16,22–25]. To investigate this, we imaged single-cell calcium activity using GCAMP6S in layers 2/3 of S1 and S2 in

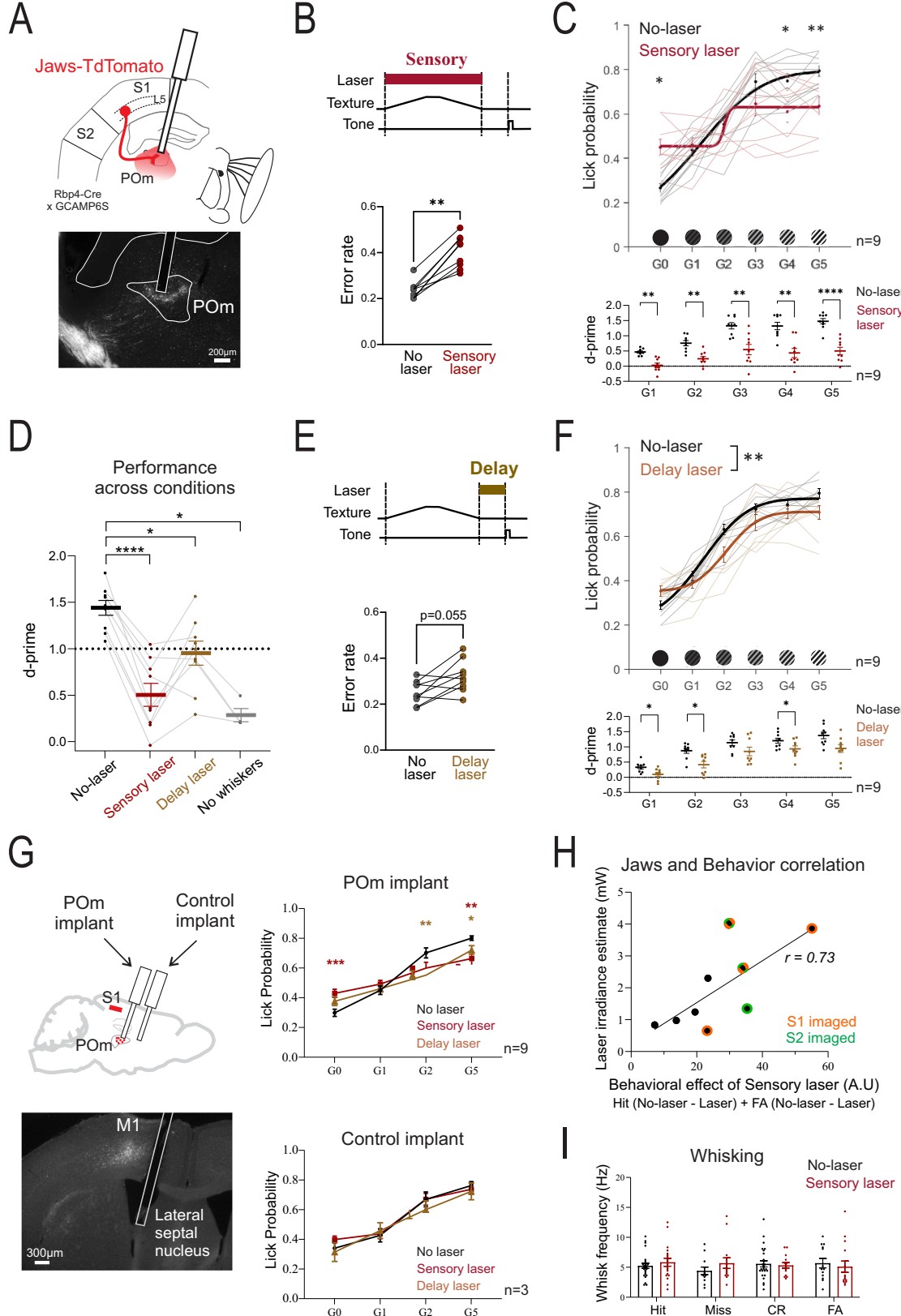

5 of the 9 Jaws-expressing mice used in the behavioral experiments (Fig. 4A).

We limited our analysis to texture-responsive cells (see "Methods" section) and quantified responsiveness by measuring the area under the curve (AUC) of the calcium transient (ΔF/F). A response was defined as the AUC during the sensory period minus that of the pre-stimulus baseline period (Fig. 4B). In contrast to silencing all layers of the cortex, which reduces cortical responsiveness to a sensory stimulus[26,27], inhibiting the transthalamic pathway did not affect overall cortical activity in response to texture presentation (Fig. 4C).

**Fig. 2 | Inactivating S1 L5 to POm impairs texture discrimination. A** Schematic and example Jaws-tdTomato in S1 L5 to POm terminals. **B** Sensory laser application and effect on error rate (average Miss and FA rate for G5 vs G0, p = 0.0039, two-sided paired samples Wilcoxon test, 9 mice). **C** (Top) Psychometric curves for no-laser (black) vs sensory laser (red) trials (G0: p = 0.0078, G4: p = 0.017, G5: p = 0.0016). (Bottom) D-prime for no-laser vs sensory laser trials (G1: p = 0.0012, G2: p = 0.0065, G3: p = 0.0012, G4: p = 0.0015, G5: p < 0.0001) (two-way RM ANOVA, Bonferroni post-hoc test, same 9 mice, 4–9 sessions each). **D** D-prime across trial conditions (no-laser vs sensory: p = 0.0091, no-laser vs delay: p = 0.08, no-laser vs no-whisker: p = 0.021, mixed effects model, Bonferroni post-hoc, 9 mice, 4 of which underwent no-whisker testing). **E** Delay laser application and effect on error rate (p = 0.055, two-sided paired samples Wilcoxon test, same 9 mice as sensory laser experiments, 4–9 sessions each). **F** (Top) Psychometric curves for no-laser (black) vs delay laser (brown) trials (effect of laser p = 0.0098, two-way RM ANOVA). (Bottom) D-prime for no-laser vs delay laser trials (G1: p = 0.023, G2: p = 0.048, G3:

p = 0.10, G4: p = 0.029, G5: p = 0.11, Bonferroni post-hoc tests, same 9 mice in **E**). **G** (Top left) Schematic of control optic fiber implanted away from Jaws expression. (Bottom left) Example control implant. (Top right) Sensory laser (red), delay laser (brown), and no-laser (black) trials for POm implant testing (no-laser vs sensory laser for G5: p = 0.0086, G0: p = 0.0007, no-laser vs delay laser for G5: p = 0.030, G2: p = 0.0047, G0: p = 0.15, 9 mice, 5–9 sessions each). (Bottom right) Same testing with control implant (3 mice, 3 sessions each). **H** Behavioral effect of sensory laser vs estimated Jaws terminal activation (see Methods section) (r: 0.73, p = 0.030, Pearson's correlation). Mice imaged in S1 (orange) and S2 (green) are shown (see Fig. 4). **I** Whisking frequency during the sensory epoch (effect of laser p = 0.58, effect of trial p = 0.91, two-way ANOVA, 4 sessions from 2 mice). Also, see Supplementary Fig. 4. Values shown as mean ± standard error of the mean. *p < 0.05, **p < 0.01, ***p < 0.001. Source data are provided as a Source Data file. Brain sections traced from the Allen Mouse Brain Atlas, https://atlas.brain-map.org/[58].

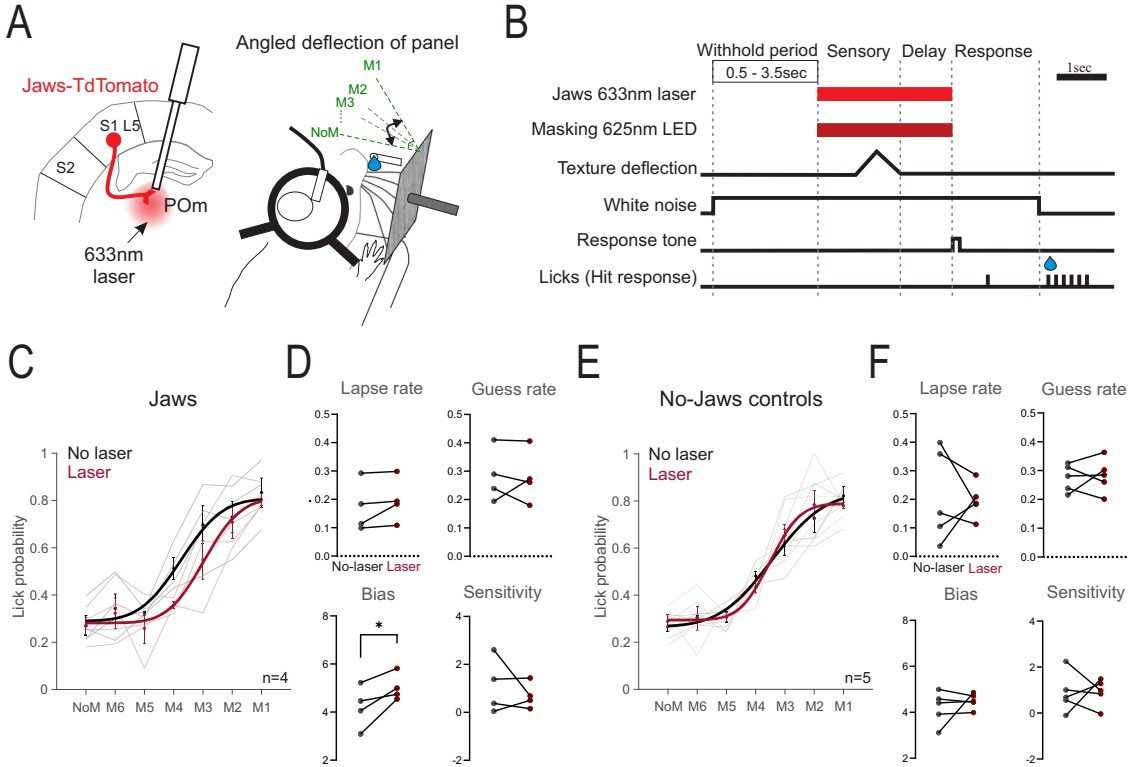

**Fig. 3 | Inhibiting the S1 L5 projection to POm impairs whisker-based detection. A** S1 L5 to POm terminals were targeted for inhibition using the Jaws opsin in Rbp4-Cre mice. Mice were trained to lick at angled movements of a panel located within their whisker field (hit) and withhold licking to no-movement (NoM), the correct rejection. **B** Time course of a hit trial. A variable pre-movement period reduced prediction of the movement, white noise masked auditory cues, and a 625 nm LED on every trial habituated the retina to red light. Movement angles M1–M6 corresponded to 20.5, 13.7, 6.9, 3.4, 1.7, 0.89°. **C** Psychometric fits for performance under no-laser (black) and laser (red) trials in mice with Jaws expression (n = 4 mice, 3 also learned the discrimination task). Datapoints for each mouse are an average of 4–6 sessions. Lines indicate data from individual mice. **D** Psychometric

parameters for the curves in (**C**). No-laser (black) vs laser (red) comparisons for lapse rate: p = 0.22, guess rate: p = 0.91, bias: p = 0.048, and sensitivity: p = 0.49 (two-sided paired t-tests). **E** Same as C but for no-Jaws control mice (n = 5: 3 TdTomato-expressing mice, 1 of which also learned the discrimination task, and 2 Jaws mice with control implants, which also learned discrimination). Datapoints for each mouse are an average of 3–6 sessions. Lines indicate data from individual mice. **F** Psychometric parameters for curves in (**E**). No-laser (black) vs laser (red) comparisons for lapse rate: p = 0.83, guess rate: p = 0.77, bias: p = 0.47, and sensitivity: p = 0.96 (two-sided paired t-tests). Error bars show the standard error of the mean. *p < 0.05. Source data are provided as a Source Data file. Brain sections traced from the Allen Mouse Brain Atlas, https://atlas.brain-map.org/[58].

In accordance with previous studies on texture discrimination[16,22–24], cells in both S1 and S2 during the no-laser condition showed higher texture responses on G5 hit trials compared to G0 CR trials (Fig. 4B, D). Laser inhibition during the sensory period did not affect this preference for the G5-rewarded texture in S1 but disrupted the discrimination in S2 (Fig. 4D, E). Hit compared to FA trials were also affected by the laser in S2 but not S1 (Fig. 4E)

These data suggest that the transthalamic pathway disrupts the differential response to texture stimuli in S2, and this aligns with

disrupted behavioral performance. To investigate the effect of laser inhibition on the link between neural response and behavior, we quantified the contributions of texture presentation and lick choice to individual neuron responses using a linear regression model (Supplementary Fig. 6). During inhibition trials, the relationship between neural responsiveness and the texture presented was inversed in S2 cells, such that a higher calcium response was associated with the CR texture (Supplementary Fig. 6B). There were no significant effects of the laser on slope coefficients in S1.

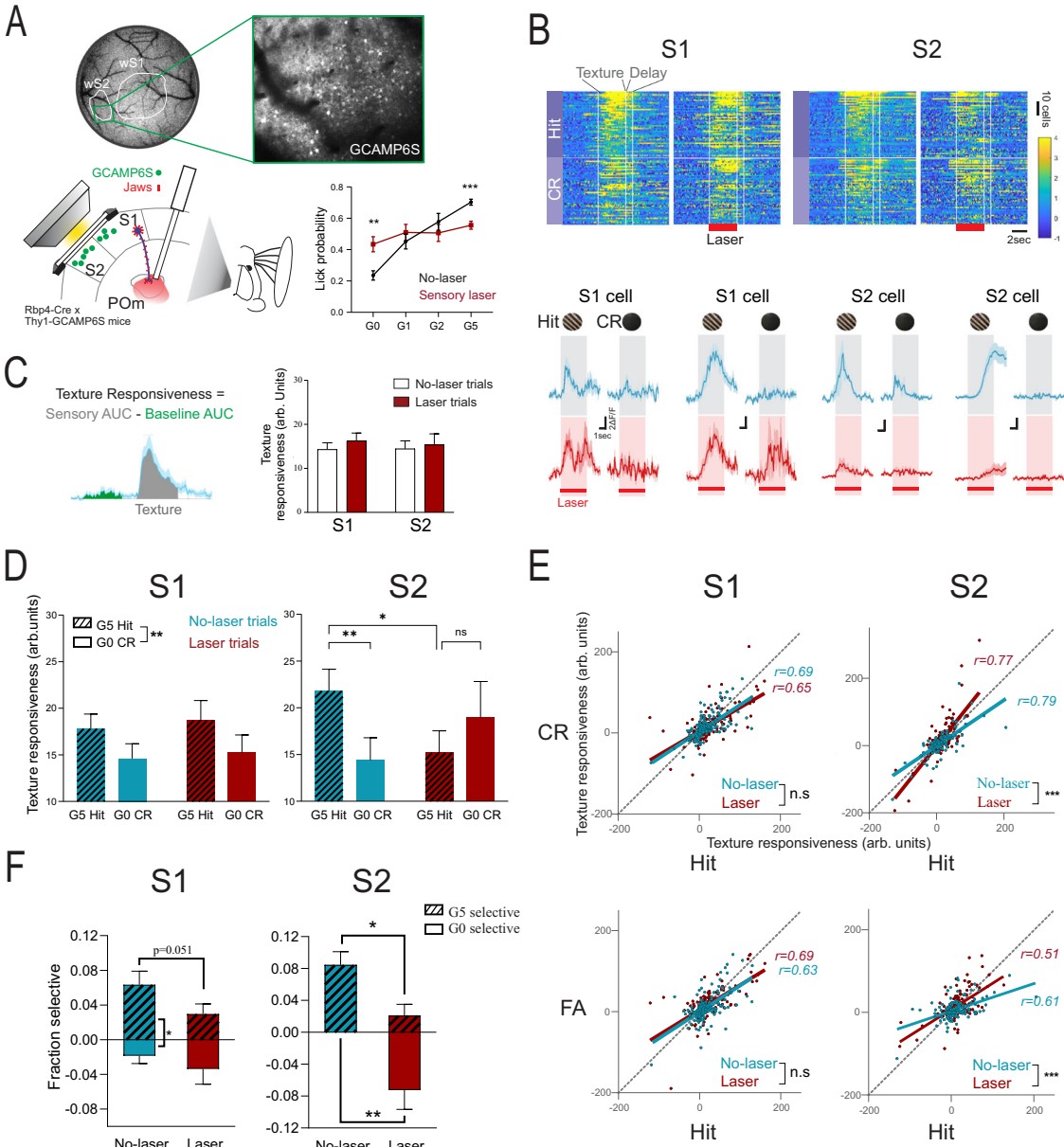

**Fig. 4 | Inhibiting S1 L5 to POm disrupts discriminability in the somatosensory cortex. A** Schematic of calcium imaging during optogenetics and (right) behavioral performance during sensory laser trials (red) (2-way RM ANOVA with Bonferroni post-hoc test, G5 no-laser vs laser: p = 0.0002 and G0 no-laser vs laser: 0.0071, n = 13 sessions). **B** (Top) Heatmaps of texture-responsive cells during Hit and CR trials. (Bottom) Example averaged fluorescence traces. Shading represents the texture period. **C** (Left) Calculation of texture responsiveness value. (Right) Texture responsiveness (arbitrary units, arb. units) for averaged G5 and G0 trials in S1 and S2 (effect of laser p = 0.16, 2-way RM ANOVA, n = 240 S1 cells from 7 experiments in 4 mice, n = 158 S2 cells from 6 experiments in 3 mice). **D** Texture responsiveness for hit (pinstripe) and CR (no-stripes) trials under no-laser (blue) and laser (red) conditions in S1 and S2. (S1: effect of texture p = 0.0033, n = 240 cells. S2: no-laser hit vs CR p = 0.0013, no-laser hit vs laser hit p = 0.032, laser hit vs CR p = 0.86, n = 158 cells, Bonferroni post-hoc test after 2-way RM ANOVA). **E** (Top) Scatterplot of texture responsiveness values for hit vs CR trials in S1 and S2 for no-

laser (blue) and laser (red) trials (simple linear regression two-tailed test, S1 no-laser vs laser: p = 0.24, n = 240 cells, S2 no-laser vs laser: p < 0.0001, n = 158 cells). (Bottom) Same scatterplot but for hit vs FA trials (simple linear regression two-tailed test, S1 no-laser vs laser: p = 0.59, n = 240 cells, S2 no-laser vs laser: p = 0.0002, n = 158 cells). **F** Fraction of cells with significant discrimination index (DI), selective for G5 (shaded) and G0 (unshaded) textures. S1: no-laser G5 vs no-laser G0 p = 0.0164, laser G5 vs laser G0 p = 0.27 (McNemar's test), no-laser G5 vs laser G5 p = 0.051 (two-tailed z-test). S2: no-laser G5 vs no-laser G0 p = 0.0009, laser G5 vs laser G0 p = 0.18 (McNemar's test), no-laser G5 vs laser G5 p = 0.025 and no-laser G0 vs laser G0 p = 0.0013 (two-tailed z-test). S1: 7 experiments in 4 mice, S2: 6 experiments in 3 mice. Values shown as mean ± standard error of the mean. *p < 0.05, **p < 0.01, ***p < 0.001. Source data are provided as a Source Data file. Brain sections traced from the Allen Mouse Brain Atlas, https://atlas.brain-map.org/[58].

To further quantify stimulus selectivity, we used receiver-operating characteristic (ROC) curve analysis to calculate a Discrimination Index (DI) for each cell, which expresses the likelihood that an ideal observer correctly classifies trial type (i.e. hit texture vs CR texture), based on the ΔF/F during the sensory period (Fig. 4F)[16,24]. A permutation test was used to determine if the DI was statistically

significant. A positive DI represents selectivity (larger response) for the G5 hit texture and a negative DI represents selectivity for the G0 CR texture. The proportion of cells with significant DI (positive or negative) was no different during no-laser and laser conditions, in S1 (no-laser 0.082 vs laser 0.063, p = 0.62, Wilcoxon test) or S2 (no-laser 0.084 vs laser 0.080, p = 0.87, Wilcoxon test). It was only under

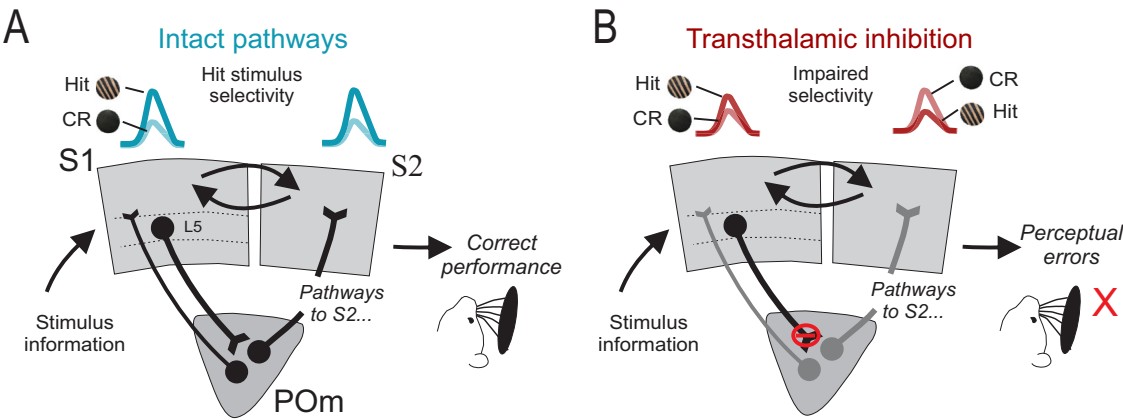

**Fig. 5 | Summary of results.** Schematic of reward texture selectivity in S1 and S2 cells and behavioral performance under conditions of (**A**) intact circuitry and (**B**) inhibition of the S1 L5 to POm projection.

no-laser conditions that there was a higher fraction of cells selective for the hit texture compared to the CR texture, in both S1 and S2 (Fig. 4F). Laser inhibition disrupted this selectivity in S2, leading to a reversal of texture selectivity preference. Hit selectivity was also reduced in S1 bordering statistical significance (p = 0.051, no-laser vs laser G5 selective fraction) (Fig. 4F). Inhibition of the S1 L5 to POm projection thus impaired reward-relevant responsiveness to textures, which is necessary for successful discrimination. This was particularly the case for cells in S2.

## Discussion

Recent evidence suggests that feedforward cortico-thalamo-cortical pathways are distinct and influential routes of sensory processing, compared to direct corticocortical projections[12], but many details of their contribution are unknown. To study this further, we inhibited the somatosensory transthalamic pathway at the terminals of S1 L5 to POm whilst monitoring neural activity in S1 and S2, and we found that this had deleterious effects on the ability of the mouse to discriminate and detect objects by whisking. These perceptual errors coincided with impaired stimulus selectivity of neurons in S2 and to a lesser extent in S1, suggesting a previously unrecognized role of cortico-thalamo-cortical pathways in sensory choice (Fig. 5). These causative results extend previous work on the neural basis of texture discrimination and cortical information flow for perceptual decisions.

### Behavioral effects

The design of our behavioral task allowed us to distinguish the effects of inhibiting the transthalamic pathway(s) separately during the sensory sampling period and during a delay period (Fig. 1), rather than silencing throughout the entire behavioral trial[14,15,17]. We report discrimination deficits due to the S1 L5 to POm inhibition during both periods, albeit to a lesser degree during the delay epoch.

**During the sensory period.** Inhibition during texture presentation severely impaired discrimination performance, increasing total errors, lapse rates and guess rates (Fig. 2B, C, Supplementary Fig. 5). The relatively large impairment due to inhibition of the S1 L5 to POm input was also reported by others when comparing the negligible effects of inhibiting either M1 or brainstem SpV inputs to POm[17].

We found deficits due to inhibition of the transthalamic pathway not only on somatosensory discrimination but also on detection (Fig. 3). Evidence exists that whisker-based detection can occur in the absence of barrel cortex[15,21] and thus presumably in the absence of the S1 L5 to POm projection. However, others show that S1 contributes to whisker-based detection[28,29] and that suppressing the S1 L5 projection to POm using chemogenetics impairs tactile detection[14]. Our results

suggest that such inhibition impairs performance by increasing the threshold to detect a small movement but does not alter detection of larger ones (Fig. 3C, D). Thus, the S1 L5 to POm projection affects detection thresholds rather than the ability to detect.

**During the delay period.** The behavioral impairment during inhibition of the delay period suggests that transthalamic signaling is ongoing after sensory sampling and contributes to task performance. Interestingly, persistent activity in cortical areas relies on constant input from the higher-order thalamus for processes such as working memory and premotor planning[30–32]. Thus, a simple explanation for this result is that inhibition of the transthalamic pathway, by reducing the input from POm, prevents the establishment of persistent activity in S2. However, it is also plausible that inhibition of this transthalamic processing reduces thalamic input to other cortical regions that may also be involved in this behavior: such potential cortical targets include association cortices such as M2[23,25], which has an S1 L5 transthalamic input[33].

By restricting inhibition to the sensory or delay epochs of the task, we avoided any potential confounds of suppressing the ability to report the sensory choice. We also confirmed that optogenetic inhibition did not affect gross whisking ability (i.e. sensory sampling) (Fig. 2I, Supplementary Fig. 4), and during the detection task, inhibition did not change perceived whisker movements during catch trials (Fig. 3C). It is thus unlikely that reduced or aberrant gross motor activity can explain the impaired behavioral performance. However, during the discrimination task, textures were presented close to the whisker pad such that active whisking was not required to sample the stimulus. Thus, we cannot rule out that inhibiting the pathway may change whisker movements during active whisking tasks, which would be predicted to be the case for an efference copy function of the transthalamic pathway[7]. Body and facial movements are intrinsically linked to sensation and perception[34], which we did not investigate.

### Effects on neuronal responses

We conclude from the behavioral data that the S1 to POm projection is crucial for somatosensory discrimination, particularly during the stimulus sampling period. S1 and S2 are the cortical origins of perceptual activity during sensory decisions[29,35–37]. This leads to the question: What information is the pathway propagating during this sensory period in these brain regions that is necessary for performance?

**Disrupted texture discriminability.** We investigated this question using 2-photon calcium imaging and concurrent optogenetic inhibition in a subset of the same mice that underwent behavioral testing. In agreement with previous studies, during texture discrimination

performance, cells in S1 and S2 show hit/CR discrimination: calcium activity in response to textures during hit trials is higher compared to correct rejection trials[18,23,25,38]. This selectivity for the hit texture over the unrewarded CR texture is a key correlate of expert discrimination performance and successful reversal learning[16,22,24]. We now show evidence that the stimulus texture selectivity is a neural substrate of correct discrimination. Inhibition of the transthalamic projection abolished hit selectivity of the S2 population (Fig. 4B, C, F), in alignment with increased performance errors (Figs. 2B, C and 4A). However, inhibition did not change the total proportion of texture-discriminating cells; instead, it changed the relative proportions of cells selective for either the hit and CR texture (Fig. 4F). In S1, cells that showed discriminability during transthalamic inhibition, were equal fractions of those selective for the hit and CR textures (Fig. 4F). This equal fraction of hit vs CR selective cells was also found in naïve mice who had not yet learned the task[16,24]. In S2 however, inhibition induced more cells to be selective for the CR texture compared to hit texture (Fig. 4F), causing a reversal of S2 population texture selectivity. Regression modeling further showed that inhibition inversed the relationship between texture and neural response (Supplementary Fig. 6B). Together, these data suggest that without a neural preference for the salient, behaviorally relevant reward stimulus, correct behavioral performance is no longer supported (Fig. 5).

**Hypothesized pathway encoding stimulus discriminability.** During multiple days of initial task learning or reversal learning, mechanisms of plasticity and reorganization of connectivity have been implicated in supporting changes in discriminability[16,39,40]. However, optogenetic inhibition occurs on a trial-by-trial basis and the effect of inhibition on population discriminability may be better explained by top-down modulation over shorter timescales[39]. For example, stimulus discriminability may depend on integrating sensory and reward inputs in S1, which are then encoded by its L5 outputs. As such, reward signals have been demonstrated in the dendrites of S1 L5 cells[41], the inhibition of which blocks reward-based learning[42]. Thus, the transthalamic pathway may deliver trial-by-trial information on the salience of stimuli to POm for subsequent propagation to S1 and the higher-order cortex. POm cells are indeed capable of differentially responding to stimuli based on reward contingencies[43]. This transmission of cortical information through POm may allow integration with subcortical information and inhibitory control[44–46].

**Differential effects of inhibition on S1 and S2.** A potential explanation for the smaller effects in S1 cells is that the optic fiber, and thus the inhibition, targeted the POm cells that project to S2, and not to S1. However, data from single-cell tracing suggest that a POm cell that projects to S2 also branches to innervate S1[47,48]. Another consideration is that we report the effects of inhibition on cells in layers 2/3, where calcium imaging was performed, whereas POm axons mainly target layer 5A layer 1 of S1, and layer 4 of S2. However, the location of presynaptic terminations is not necessarily where the postsynaptic cell body resides, and indeed, neurons in layer 2/3 of S1 receive a major monosynaptic input from POm[49,50], despite their terminations avoiding this layer[51]. Thus, the transthalamic pathway innervates layer 2/3 cells in S1, but inhibiting the pathway did not have as robust effects on these cells, compared to layer 2/3 cells in S2. This result makes sense in a hierarchical framework, where the POm to S1 pathway is considered feedback and that from POm to S2 is considered feedforward[5]. It also makes sense at a synaptic level, where POm to S1 glutamatergic projections have synaptic properties that modulate the postsynaptic cell[50], whereas POm to S2 synapses are all fast, robust, all-or-none projections, termed 'driver' projections[5]. Driver projections are those that deliver stimulus information[6,8,9]

**Some provisos.** We have demonstrated that the S1 L5 input to POm is required for processing sensory discrimination, thus implicating the first leg of transthalamic processing in propagating information relevant to this function to the cortex. Our reported effects of inhibition in layer 2/3 of the cortex may indeed be monosynaptic from POm[5,50], but may also involve multiple synapses, such as via deeper layers of S1 or S2 before reaching L2/3. We also appreciate that other brain regions between POm and S1 or S2 are involved, such as a transthalamic pathway from S1 L5 to M1 via POm[33]. However, projections from POm to S2 are the most likely direct candidate: evidence of a strong S1 to POm to S2 transthalamic pathway exists[5,10] and during whisker-based tasks, inactivation of S2, but not M1, impaired performance[28]. In any case, the inclusion of other brain regions still involves transthalamic processing, which is the main conclusion of our study.

Finally, our experimental design involved using transgenic mice in which Cre in the cortex is limited to L5 cells (Rbp4). We did this in order to place Jaws in such cells that innervate POm without involving the L6 projection there. However, it appears that not all L5 cells in Rbp4 mice contain Cre[52], and so it follows that not all L5 to POm inputs would be inhibited by Jaws. Furthermore, the complete efficiency of Jaws expression in L5 cells containing Cre is questionable, and precisely-targeted probe placements are required for strong behavioral and imaging effects (Fig. 2H, Supplementary Fig. 5). For all of these reasons, we have almost certainly *underestimated* the effects of inhibiting the transthalamic pathway.

## Direct corticocortical versus transthalamic pathways in perceptual tasks

Cortical inhibition experiments have shown that S1 and S2 are critical for somatosensory decisions[15,21,29]. However, it is important to note that these silencing strategies also inhibit transthalamic pathways through cortical L5. We show that the transthalamic pathway is not necessary for detection but does affect its threshold (Fig. 3) and appears essential for discrimination ability, suggesting a more complex role of the pathway rather than simple propagation of the presence of a sensory stimulus. Indeed, transthalamic inhibition neither acted to reduce overall neural activity nor changed the fraction of texture-discriminating cells in the cortex. The impact of inhibition was specific to the neural activity relevant to correct behavior, and our data suggest that one of the roles of the transthalamic pathway is the delivery of reward-related stimulus information to S2 for correct sensory discrimination. As mentioned, the transthalamic pathway at its thalamic node may also act to integrate subcortical information and cortical reward information relevant to task performance.

The severe deficits in discrimination performance we report here from corticothalamic inhibition occurred in the presence of functional corticocortical pathways. So what then is the role of the direct corticocortical pathway? Studies that compare the role of corticocortical projections and corticothalamic projections (presumably involved in transthalamic processing) show that direct corticocortical pathways make minor contributions to sensory decisions and it is the corticothalamic pathways that substantially support perceptual performance[11,14]. Our results thus add to the mounting evidence suggesting that transthalamic signaling plays a relatively major role in perceptual processing in the cortex. However, the impacts of direct corticocortical and transthalamic signaling have yet to be directly compared, and open questions remain, such as detailed motor contributions, the role of the transthalamic pathway in linking other cortical regions, and during other tasks.

In summary, we have presented behavioral and neuronal data that implicate the transthalamic pathway in propagating stimulus feature selectivity for correct perceptual decisions. Our results demonstrate the need to include transthalamic circuits and higher-order thalamus in the framework for cortical functioning in perception.

## Methods

### Animals

All experiments were performed in accordance with protocols approved by the Institutional Animal Care and Use Committee at the University of Chicago. Transgenic mice expressing Cre-recombinase in L5 of cortex (Rbp4-Cre)[53] were bred by crossing hemizygous male Tg(Rbp4-Cre) KL100GSat/Mmcd mice (GENSAT RP24-285K21) with female C57Black6J mice. Rbp4-Cre positive offspring were used for in vivo electrophysiology experiments (Supplementary Fig. 2). All other data were generated with Rbp4-Cre x Thy1-GCAMP6S mice, created by breeding male Rbp4-Cre mice with female Thy1-GCaMP6S mice (GP4.12Dkim/J, Stock: 025776, The Jackson Laboratory). Tail biopsies were taken at 14-21 days old and genotyped by real-time polymerase chain reaction (Transnetyx, Cordova, TN). Mice used in behavioral experiments were a balanced mix of male and female and housed individually on a reverse light-dark cycle (7am–7pm). All mice were given food and water *ad libitum*, unless water restricted as described.

### Surgical procedures

Mice were anesthetized with ketamine (100 mg/kg)/xylazine (3 mg/kg, i.p) and maintained on isoflurane (1.0–1.5% in oxygen). A small burr hole was made over the target site and the virus was injected using a 0.5 μL syringe (7000.5 KH, Hamilton) at a rate of 5–10 nl/min. To express Jaws-TdTomato or TdTomato alone in L5 of S1, 250 nl of AAV8-CAG-FLEX-Jaws-KGC-TdTomato-ER2 (UNC Vector Core) or AAV5-CAG-FLEX-TdTomato (UNC Vector Core) was injected into left S1 of 6-week old mice. S1 coordinates relative to bregma were DV: −0.5, ML: 3.2 mm, AP: −1.3 mm. After 2 weeks, a custom titanium head post (11.5 mm diameter, H.E. Parmer) was adhered to the skull using dental cement (C&B Metabond). In the same surgery, a 4 mm circular cranial window (0.66 mm thick, Tower Optical) attached to a 5 mm glass coverslip was implanted over S1 and S2 (DV: −0.5, ML: 4.2 mm, AP: −1.2 mm). An optic fiber stub (200 μm diameter, 0.5NA, Thorlabs) was also implanted at 8° from the vertical to target left POm (DV: −3.15 mm, ML: 0.85 mm, AP: −1.25 mm). In three mice, a second optic fiber was implanted sub-cortically, anterior to POm after behavioral training. After all surgeries, analgesia (Meloxicam, 1–2 mg/kg, s.c) was administered pre-operatively and 24 h post-operatively.

### Intrinsic signal optical imaging

S1 and S2 were functionally located in each mouse using intrinsic signal optical (IS) imaging. Mice were anesthetized with 3% isoflurane in oxygen and maintained on 1% isoflurane. Reflected light through the cortical window was imaged using a CCD camera (Teledyne QImaging, Retiga-SRV). The surface vasculature was visualized under green illumination (525 nm) and hemodynamic response was captured under red illumination (625 nm). Multi-whisker responses were stimulated with a textured panel (3 Hz antero-posteriorly, 2 cm from the whisker pad). Single whiskers were threaded with a pipette tip for stimulation (5 Hz antero-posteriorly). Images were acquired with custom MATLAB code 1 s after stimulus application (4 s duration), and alternated with no-stimulation trials (30 trials each, 8 s inter-trial interval). The signal was quantified as the difference in the reflected light during the stimulus trials and no-stimulus trials.

**Trimming to task-relevant whiskers.** For each mouse, the whisker map generated from IS imaging was overlaid with the fluorescence image of Jaws-TdTomato. If the S1 barrels of task-relevant whiskers did not express Jaws-TdTomato, the corresponding whiskers were trimmed down to the whisker pad (Fig. 1C). Task-relevant whiskers were verified for each mouse as those that consistently contacted the texture panel during the discrimination task (typically alpha, A1, beta, B1, B2, gamma, C1, C2, delta, D1, and D2 whiskers). Whisker length was monitored every 3–4 days and re-trimmed as necessary.

### Behavioral setup and discrimination task

**Setup.** The behavioral enclosure was lightproof and fitted with soundproof panels (0.8 NRC, Sound Seal) with the following internal light sources: An infrared webcam (webcamera_usb) and a 625 nm LED (Thorlabs) with the output diffused with an acrylic panel and Kimwipe tissues. The 625 nm LED was positioned 20 cm in front of the mouse's face and emitted during the sensory period of every trial. Speakers were positioned so the response tone (8 kHz, MATLAB) was 60 dB at the distance of the mouse. A pump (NE-1000 syringe pump, New Era) delivered water through a spout (15G blunt needle) mounted 3–6 mm away from the mouth. Licks were detected by an attached capacitance sensor (Teensy 3.2, PJRC). Mice were able to run freely on a custom-built treadmill.

Textured panels (5.5 cm diameter circles) were attached to a custom-built 8-sided wheel (7 cm radius), rotated by a stepper motor (X-NMS17C, Zaber). Textures were presented one at a time to the mouse by mounting the stepper motor vertically on a linear slider (X-LSM050, Zaber) and advancing it into, and retracting it out of, the right whisker field. At the start of each trial, the wheel was rotated in either direction for a random amount of time (0.34–1.8 s) to prevent predictive auditory cues. The linear slider advanced the texture from 3.7 cm from the whisker pad, to 2.2 cm away, still out of reach of most whiskers, and then into the whisker field, 1.2 cm away from the pad. The texture was held at this position for 0.5 s and retracted, upon which there was a delay period (0.6–1 s), followed by the tone and response period. Synchronization and triggers were controlled through MATLAB (2020b).

**Habituation.** One week after window surgery, mice were water restricted to 80–95% weight and habituated to the head-fix apparatus within the enclosure over 3–5 days. During the habituation period, mice were first encouraged to lick the spout (8 μL water delivery). Then, water delivery was paired with a preceding tone (8 kHz) until mice self-triggered the delivery by licking the spout within 2 s of the tone onset.

**G5 and G0 discrimination training.** Mice were then trained on a go/no-go design to discriminate between a smooth texture (no-go: matte, black, aluminum foil) and one with a grating (go: gratings on foil). The grating textures were made of 5 mm-wide sandpaper strips, 7 mm apart on black foil. During initial training, the presentation of a grating texture (P20 grit strips: G5 texture) was associated with a water reward (6 μL). For the first few G5 trials, water delivery was triggered immediately after the response tone. Then, water was only delivered if mice licked the spout after the tone, but within the response period (1.8 s). Water delivery only occurred after the end of the response period. Licking during the delay period triggered one alarm beep (NE-1000 syringe pump, New Era), aborting the trial for immediate restart.

If the mouse licked during the response period of a G5 trial, the trial was deemed a "hit". No-lick response to the G5 texture during the response period was regarded a "miss". If the mouse performed 3 hit trials in a row, then the no-go G0 texture was introduced. Licking during the response period to the G0 texture triggered a beeping alarm (65 dB, NE-1000 syringe pump) and 12 s timeout ("false alarm", FA). If mice correctly withheld a lick response to the G0 texture, this was a "correct rejection" (CR) and punishment was avoided. If the previous two G0 textures were FAs, a mild air puff to the snout region was also administered (Cleaning duster, Office Depot). The consequence for a miss and CR response was moving to the next trial. G5 and G0 textures were presented in random order but with not more than 2 in a row and the inter-trial interval was 3–5 s.

During training, the 0.6 s delay period was extended to 1 s and the response period was shortened from 2 to 1.6 s. Mice were considered trained when performance reached a d-prime of >1 or >70% correct for two consecutive days. On training and testing days, mice were allowed

to perform the task until sated and then supplemented in their home cage with 0–0.8 ml water to maintain a body weight of 80–95%.

**Psychometric and laser testing.** After learning to discriminate G5 and G0, textures of various coarseness were presented in randomized order (G4 = P100, G3 = P220, G2 = P1500, G1 = foil strips on foil) in addition to G5 and G0. G0 was presented on 30% of trials and all grating textures (G1–G5) were rewarded. Effects of the laser (see "Optogenetic inhibition" section) were initially tested during two conditions, in separate testing sessions on alternate days: (1) laser during the sensory period (3.8 s); (2) laser during the delay period (1 s). For each session, laser and no-laser trials were presented at 50% each, except when the mouse was returned to G0 and G5 training (see "Bias correction" section). In a third testing condition, both sensory laser trials and delay laser trials were tested in the same session but only textures G0, G1, G2, and G5 were presented. This testing was used during 2-photon imaging. Behavioral sessions were run daily and typically consisted of 150–400 trials lasting between 1–2 h. Only sessions during which mice performed for no-laser trials (d-prime > 1 or > 70% correct) were included in the analyses.

**Bias correction.** To combat the high bias towards go responses during the beginning of each session[54], mice are minimally water restricted where possible (80–95%) and given "warm-up" training on G5 vs G0 discrimination. To proceed to psychometric testing, mice must respond to randomized G0 presentations with three CRs in a row (no FAs), which typically required 7-150 trials. Mice are returned to G5 vs G0 training if they perseverate (lick at every trial) or disengage (no-lick response) on a sliding window of 7 trials. Any G5 v G0 bias correction trials were excluded from analyses.

**Behavioral analyses.** Correct performance was calculated for each session from G5 and G0 trials, by the formula (hit trials + CR trials) / (hit + CR + miss + FA trials) × 100.

D-prime was calculated by the difference in z transforms of the G5 hit rate and FA rate:

$$d' = z(\text{hit}) - z(\text{FA}) \qquad (1)$$

Error rate was calculated by taking the average of mean miss trials during the G5 presentation and mean FA trials. For each laser testing condition in each mouse, psychometric curves were fitted with a 4-parameter sigmoidal cumulative Gaussian function[55]:

$$y(x) = g + (1 - g - l) * 0.5 * (1 + \text{erf}((x - u)/\text{sqrt}(2 * v^{\wedge}2))) \qquad (2)$$

where y(x) is the lick probability, x is the texture and erf represents the error function. The parameters to be fitted are: g (guess rate), l (lapse rate), u (subject bias), and v (discrimination sensitivity).

To plot the relationship between the behavioral effect and Jaws activation for each mouse (Fig. 2H), the additive difference between the guess and lapse rates during no-laser and sensory laser trials was plotted against an estimation of laser irradiance, relative to Jaws terminal activation. This laser irradiance was calculated by multiplying the area of Jaws terminal expression in the brain section with the deepest implant location, by the estimated irradiance loss from the tip of the implant to the middle of the expression site (https://web.stanford.edu/group/dlab/cgi-bin/graph/chart.php, 630 nm, 0.5NA, 4 mW, 0.1 mm fiber radius).

**Optogenetic Inhibition**
Optic fiber implants (200 μm diameter, 0.5NA, Thorlabs) were attached to a patch cable (0.5NA, Plexon) to deliver a 633 nm laser (LuxX 633-100, Omicron-Laserage) with an estimated power output of 4 ± 0.2 mW. To reduce fluorescence artifacts of the laser during

2-photon imaging, autofluorescence from the patch cable was removed by photobleaching overnight using a 620 nm LED (PlexBright, Plexon, 3.5–5 mW at tip). During imaging, the red channel photomultiplier tube (PMT) was shut closed.

**Whisker tracking**
Whisker movements were recorded from above at 122fps using a CMOS high-speed camera (Basler acA800-510 μm) and infrared lens (6 mm C VIS-NIR Series, Edmund Optics). The whiskers were illuminated with light from the IR camera and white cardboard provided background contrast. However, light levels were not increased for optimal tracking to avoid interfering with concurrent calcium imaging. Thus, whiskers could only be successfully tracked for a subset of sessions. We modified the available code (https://github.com/jvoigts/whisker_tracking) based on a convolutional neural network to label whiskers and a Hough transform to extract whisker positions and angles (https://github.com/sdemyanov/ConvNet). Analysis was restricted to movements from whiskers within 2 cm of the whisker pad during the texture presentation period of the task for G5 (hit and miss) and G0 (CR and FA) trials. A whisk was defined as a continuous sweep in the antero-posterior axis. The rate of whisking was calculated by the number of whisks / sensory period (3.84 s). See also Supplementary Fig. 4.

**Detection task**
In the same behavioral setup as the discrimination task, mice were trained to detect the movement of a textured panel (P20 sandpaper) positioned within the whisker field (1.5 cm from the whisker pad). The panel was mounted to the same texture wheel used in the discrimination task. The panel was deflected anteriorly and immediately returned to its original position.

The task followed a go no-go task design, but here the panel movement was the go rewarded cue and no-movement was the no-go punished cue. To minimize timed prediction of a movement, a randomized "withhold" period of 0.5–3.5 s preceded the deflection. If mice licked during this period, an alarm beep would sound (New Era) and the trial was aborted and restarted. White noise (MATLAB) throughout the trial masked auditory cues from the stepper motor. There was a 1 s delay after the sensory period (panel movement) but before the response tone (8 kHz). Licks during the delay period also aborted the trial for restart. Mice are free to move on a treadmill.

Mice were first trained to lick to the largest movement (M1: 20.5°) compared to no-movement (NoM: 0°) and considered trained when performance reached d-prime of > 1 or > 70% correct for two consecutive days. This typically requires 3–7 days. Psychometric and laser testing was then conducted with a range of angles: M2–M6 corresponding to 13.7, 6.9, 3.4, 1.7, 0.89°. A 625 nm LED was emitted during the sensory and delay periods of every trial. The 633 nm laser was applied throughout the sensory and delay periods of the task. Behavioral analyses followed that of the discrimination task.

**Two-photon calcium imaging**
GCAMP6S activity from Rbp4-Cre × Thy1-GCAMP6S mice was excited through a 16× objective (0.8NA, Zeiss) using a Ti:Sapphire laser (DeepSee, Spectra-Physics) tuned to 920 nm. PrairieView software controlled a resonant scanner of a multiphoton microscope (Ultima Investigator microscope, Bruker). Fluorescence was collected through a green emission filter (et525/70m-2p, Chroma Tech, VT, USA) and detected by a GaAsP PMT (Hamamatsu Model H10770). Image sequences over a 512 × 512 μm field of view were captured at 7.5 Hz in layer 2/3.

Each imaging session targeted S1 or S2, which were located through the cranial window based on blood vessel landmarks and whisker mapping results (see "Intrinsic signal optical imaging" section). The border area was avoided. In animals imaged more than once,

no areas overlapped and only cells imaged in one session were included in the analyses. Five mice in total were imaged in both S1 and S2. However, in 3 mice, analyses were restricted to one region each due to poor GCAMP6S signal, occlusion of the window, or not enough behavioral trials.

**Image processing.** Raw fluorescence images were pre-processed using Suite2p (https://suite2p.readthedocs.io/)[56]: regions of interest (ROIs) were selected and their fluorescence signals were extracted and deconvoluted. ROIs fluorescence traces for each experiment were visually confirmed and analyzed using custom MATLAB code. Fluorescence time-series were normalized to percent change from a time-varying baseline ($\Delta F/F$). Baseline fluorescence was estimated for each neuron by thresholding raw fluorescence to eliminate spike-induced fluorescence transients, which was thresholded and smoothed with a 4th-order, 81-point Savitzky–Golay filter[57].

**Responsive cells and area under the curve analysis.** Each imaging trial included a baseline period of 4 s before moving the texture into the whisker field. Responsive cells were defined as ROIs with a significant difference between the mean dF/F during the baseline period compared to the mean dF/F during the texture period, for any G5 and G0 texture trials, including sensory laser trials. A significant difference was determined by a Wilcoxon signed rank test ($p < 0.05$) when the absolute value skew $<0.6$ or a sign test when the absolute value skew $>0.6$. Only responsive cells were used in subsequent analysis. Due to the slow kinetics of GCAMP6S and the long sensory period of the task, we used the area under the curve (AUC) of the dF/F trace as a measure of responsivity to the texture presentation. Texture responsiveness value was calculated as the AUC during the sensory period – AUC during the baseline period. Both periods were 3.8 s long. To determine the effect of the laser on cells that reduced response to the texture, any cell that showed negative texture responsiveness values during no-laser G5 trials had their other trial responsiveness values reversed in sign (Fig. 4C).

**Single-neuron discrimination index analysis.** To quantify how well single cells could discriminate between the two hit and CR textures, a discrimination index (DI) was calculated based on neurometric functions using a receiver-operating characteristic (ROC) analysis[16,38]. The AUC of calcium signals during the stimulus presentation period minus the AUC of an equivalent baseline period in the G5 hit texture trials were compared to that of the G0 CR texture trials. ROC curves were generated by plotting, for all threshold levels, the fraction of G5 hit trials against the fraction of G0 CR trials for which the response exceeded the threshold. Threshold levels were defined as a linear function from the minimal to the maximal calcium signals. DI was computed from the area under the ROC curve by $DI = (AUC - 0.5) \times 2$. DI values vary between $-1$ and $1$. Positive values indicate a larger response, or selectivity, to the G5 hit texture compared to the G0 CR texture, whilst negative values indicate a selectivity to the G0 CR over the G5 hit texture. DI values above chance were assessed using permutation tests, from which a sampling distribution was obtained by shuffling the texture labels of the trials 1000 times. The measured DI was considered significant when it was outside of the 2.5th–97.5th percentiles interval of the sampling distribution.

**Fluorescence microscopy**
Mice were perfused with cold, phosphate-buffered saline (pH 7.2, 50 ml) followed by cold, 4% paraformaldehyde (100 ml). Brains were extracted, sucrose-protected over two days, and sectioned on a sliding microtome. Brain sections were cut 50 μm thick and mounted with Superfrost slides. Some fluorescent photos were captured before mouting. Fluorescence signals were visualized under a fluorescence microscope (Leica Microsystems) using the appropriate filter cubes.

Images were captured using a Retiga-2000 CCD monochrome camera and QCapturePro imaging software (Teledyne QImaging, Surrey, BC). Image post-processing such as estimating the distance from the optic fiber implant (Fig. 2H) was performed with ImageJ software.

**Statistics**
Statistical tests were conducted in MATLAB (2020b) or Prism software (v.9, GraphPad). A Shapiro–Wilk test (<30 samples) or Kolmogorov Smirnoff test (>30 samples) was used to test for normality. If no significant departure from normality was found, parametric tests were used. For departures from normality, the non-parametric Wilcoxon signed rank test was used for unpaired values and the Wilcoxon matched-pairs signed rank test for paired values. Where normality differed for tests within an experiment, the more conservative non-parametric test was applied across the experiment for consistency. A significance level was set at 0.05 and multiple comparisons were adjusted with the Bonferroni correction unless otherwise indicated. Specific statistical tests used and sample sizes are indicated in figure legends, text, and Supplementary Table 1.

**Reporting summary**
Further information on research design is available in the Nature Portfolio Reporting Summary linked to this article.

## Data availability
The data generated in this study are provided in the Source Data file attached to this paper. Source data are provided with this paper.

## Code availability
Data analysis scripts are publicly available at https://github.com/mobrainer/Transthalamic_2024.

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

## Acknowledgements

We thank Masaki Makitani, Adam Kunz, Jessica Manieson, and Graham Fetterman for their technical assistance, and Jackson Cone for comments on drafts of the manuscript. This work was supported by the National Institutes of Health (Grants NS094184 and EY022388 to S.M.S. and F31EY031965 to C.McK.) and the National Health and Medical Research Council, Australia (Grant 2003646 to C.M.). The Florey Institute of Neuroscience and Mental Health acknowledges the strong support from the Victorian Government and in particular the funding from the Operational Infrastructure Support Grant.

## Author contributions

C.M. and S.M.S. conceived the project and wrote the manuscript. C.M. performed the experiments, C.M. and C.McK. performed the analyses.

## Competing interests

The authors declare no competing interests.
