## [Peer Review File · Nature Communications]

A TRANSTHALAMIC PATHWAY CRUCIAL FOR PERCEPTIONEditorial Note: This manuscript has been previously reviewed at another journal that is not operating a transparent peer review scheme. This document only contains reviewer comments and rebuttal letters for versions considered at *Nature Communications*.

REVIEWERS' COMMENTS

Reviewer #2 (Remarks to the Author):

Thank you for giving me the opportunity to review this paper by Mo and colleagues, focusing on the author's response to R3's comments.

R3 raised several important points regarding the novelty and interpretability of the findings presented. As far as I understand, the major critiques are:

1. Whether the S1L5 terminal inhibition in PO goes beyond what has been already established in Qi et al., 2022
2. Whether the authors can actually claim strong evidence for a transthalamic route based on the findings presented.

From reading the authors' response to R3's comments, my impression is that the authors think that these two points are related insofar that they have measured S2 responses as a downstream target and that neurons are less sensitive to texture differences when S1L5 terminals are suppressed in PO. Also, because the authors only inactivate these inputs during the sensory period of the task, they conclude that they are the first to show evidence for a transthalamic route.

I have read the paper and think that there is a lot to like about it, as R3 had noted. However, I do not think that the authors' response addresses R3's concerns. The answers stated are tautological to a large degree-- S2 shows less discrimination but the behavior is also worse, and the only way to conclude that this is related to a transthalamic route is to assume that one exists!

Also, there are clear S1 effects (the p value is 0.051), but the authors treat this as a null effect, which is perplexing.

R3 asked for a more definitive test-- record from PO in this experiment and compare the terminal inhibition in PO to that in S2. I think these are very reasonable requests. Without those important experiments being consistent with their thesis, the authors cannot claim a transthalamic route and should really amend the title and the abstract at this point to reflect what they found--an important role for S1 inputs to PO in discrimination. As they state, the transthalamic route is one out of several possibilities that explains their effects. In my view, it can certainly continue to be an interesting discussion point at this stage and to motivate future experiments.

Lastly, and maybe a minor point, the authors criticize Qi et al for having potential motor effects, but the Qi paper clearly showed no effects of S1L5-PO inactivation on motor performance (Fig. S7).

REVIEWERS' COMMENTS

Reviewer #2 (Remarks to the Author):

Thank you for giving me the opportunity to review this paper by Mo and colleagues, focusing on the author's response to R3's comments.

R3 raised several important points regarding the novelty and interpretability of the findings presented. As far as I understand, the major critiques are:

1. Whether the S1L5 terminal inhibition in PO goes beyond what has been already established in Qi et al., 2022
2. Whether the authors can actually claim strong evidence for a transthalamic route based on the findings presented.

From reading the authors' response to R3's comments, my impression is that the authors think that these two points are related insofar that they have measured S2 responses as a downstream target and that neurons are less sensitive to texture differences when S1L5 terminals are suppressed in PO. Also, because the authors only inactivate these inputs during the sensory period of the task, they conclude that they are the first to show evidence for a transthalamic route.

I have read the paper and think that there is a lot to like about it, as R3 had noted. However, I do not think that the authors' response addresses R3's concerns. The answers stated are tautological to a large degree-- S2 shows less discrimination but the behavior is also worse, and the only way to conclude that this is related to a transthalamic route is to assume that one exists!

We appreciate Reviewer 2's concern that we have not demonstrated a transthalamic pathway insofar as direct linkage from POM to S2. However, we do not state in the manuscript that we are looking at S1L5 to POM to S2. We are investigating the cortical effects of inhibiting a corticothalamic projection – hence transthalamic. An impact on cortex, wherever it is a direct projection from POM to S2 (which is the most likely), or via some intermediary(s), is still by definition 'transthalamic'. (See accompanying email.)

We discuss this on line 376:

“We also appreciate that other brain regions between POM and S1 or S2 are involved, such as a transthalamic pathway from S1 L5 to M1 via POM (Mo and Sherman, 2019). However, projections from POM to S2 are the most likely direct candidate: evidence of a strong S1 to POM to S2 transthalamic pathway exists (Theyel et al., 2010; Miller-Hansen and Sherman, 2022) and during whisker-based tasks, inactivation of S2, but not M1, impaired performance (Le Merre et al., 2018). In any case, the inclusion of other brain regions still involves transthalamic processing, which is the main conclusion of our study.”

Also, there are clear S1 effects (the p value is 0.051), but the authors treat this as a null effect, which is perplexing.

The Reviewer is referring to **Fig. 4F**, the ROC analysis to calculate a texture discrimination index for S1 and S2 neurons. We do not state that there is a null effect of inhibition on S1.

Line 236: “Hit selectivity was also reduced in S1 bordering statistical significance ($p=0.051$, no-laser vs laser G5 selective fraction) (**Fig. 4F**). Inhibition of the S1 L5 to POM projection thus impaired reward-relevant responsiveness to textures, which is necessary for successful discrimination. This was particularly the case for cells in S2 (**Fig. 4G**).”

Perhaps the Reviewer is referring to data from **Fig. 4D** and **E**, which show no effect of inhibition on S1 cell texture responsiveness values. We stated on line 223 that “There were no significant effects of the laser on slope coefficients in S1.”

However, there is no other way to interpret this data. There is no effect on S1 cells (simple linear regression, S1 no-laser vs laser: $p=0.24$).

In the Discussion, we refer to the S1 neuronal results as impacted by the inhibition, but to a less extent compared to S2. We do not believe, and have never stated, that S1 cells are not affected by the inhibition.

Line 352: “A potential explanation for the **smaller effects** in S1 cells...”

Line 361: “layer 2/3 cells in S1, but inhibiting the pathway **did not have as robust effects** on these cells, compared to layer 2/3 cells in S2”

R3 asked for a more definitive test-- record from PO in this experiment and compare the terminal inhibition in PO to that in S2. I think these are very reasonable requests. Without those important experiments being consistent with their thesis, the authors cannot claim a transthalamic route and should really amend the title and the abstract at this point to reflect

what they found--an important role for S1 inputs to PO in discrimination. As they state, the transthalamic route is one out of several possibilities that explains their effects. In my view, it can certainly continue to be an interesting discussion point at this stage and to motivate future experiments.

Again, an impact on cortex, wherever it is a direct projection from POm to S2 (which is the most likely), or via some intermediary(s), is still by definition 'transthalamic'. Furthermore, we have already addressed this point in previous versions.

Lastly, and maybe a minor point, the authors criticize Qi et al for having potential motor effects, but the Qi paper clearly showed no effects of S1L5-PO inactivation on motor performance (Fig. S7).

Figure S7 plots how inhibition affects trial duration (B), time spent in sensory zone (C), and the intertrial interval (D). A more subtle and direct motor output (whisking, controlled by S1 L5) could have been analyzed but the Reviewer is correct, this shows that their inhibition did not affect motor response at this level of behavioral analysis.